# Acute Effects of High-Intensity Functional Training and Moderate-Intensity Continuous Training on Cognitive Functions in Young Adults

**DOI:** 10.3390/ijerph191710608

**Published:** 2022-08-25

**Authors:** Manuel de Diego-Moreno, Francisco Álvarez-Salvago, Antonio Martínez-Amat, Carmen Boquete-Pumar, Antonio Orihuela-Espejo, Agustín Aibar-Almazán, José Daniel Jiménez-García

**Affiliations:** 1Deparment of Physical Education, University of Wales, Trinity, Saint David, 29018 Málaga, Spain; 2Department of Health Sciences, Faculty of Health Sciences, University of Jaén, 23071 Jaén, Spain; 3Department of Physiotherapy, Faculty of Health Sciences, European University of Valencia, 46112 Valencia, Spain

**Keywords:** high-intensity, cognitive function, moderate-intensity, acute effect

## Abstract

Background: The purpose of the present study was to compare the influence of an acute bout of high-intensity functional training (HIFT) with an acute bout of moderate-intensity continuous training (MICT) on measures of cognitive function. Methods: Sixty-nine young adults (Mean ± SD: age *=* 21.01 ± 2.79 yrs; body mass *=* 69.65 ± 6.62 kg; height *=* 1.74 ± 0.05 m; Body Mass Index *=* 22.8 ± 1.41) gave informed consent and were randomly divided into three groups. The HIFT group, with 27 participants, performed a high-intensity (>85% Max. HR) circuit of functional exercises for 30 min. The MICT group, with 28 participants, performed moderate-intensity (70–80% Max. HR) continuous training on a cyclo-ergometer. The control group did not perform any activity. The Stroop Test, Word Recall and N-Back Test were completed to assess during the familiarization period, immediately before and immediately after the training’s bouts. Results: The repeated measures ANOVA did not show significant mean differences for any group. However, the T-Test for the paired samples demonstrated very significant differences in the Stroop Test, in terms of fastest response time (FRT; mean difference (MD) *=* −1.14, *p <* 0.01, d *=* 0.9), mean response time (MRT; MD *=* −2.16, *p <* 0.01, d *=* 0.66) and the number of correct answers (NCA; MD *=* 1.08, *p <* 0.05, d *=* 0.5) in the HIFT group and in the MICT group (FRT; MD *=* −1.79, *p <* 0.01, d *=* 0.9), (MRT; MD *=* −3.07, *p <* 0.01, d *=* 0.9) (NCA; MD *=* 1.54, *p <* 0.05, d *=* 0.5). Conclusions: There were no differences in the control group. HIFT and MICT may elicit specific influences on cognitive function, mainly in executive function and selective attention.

## 1. Introduction

Most research to date has used aerobic exercise, which is continuous moderate- or low-intensity exercise, to examine the effects of physical activity and exercise on brain function. Small-to-moderate gains were seen in areas such as processing speed, attention, executive function and memory in a number of meta-analyses involving both acute and chronic therapies [1,2,3,4,5,6,7,8,9,10,11].

According to the theory, exercise has an acute impact on cognitive function that follows an inverted U relationship, which is similar to the arousal theory first put forth by Yerkes and Dodson [12]. These researchers were the first to postulate that, as exercise intensity increases, cognitive function gets better until a critical intensity is reached, at which it would become impaired. While studies examining the influence of high-intensity or overloaded exercise report ambiguous results [13,14], demonstrating both positive [15,16,17] and negative [13,14] results on cognitive function, scientific evidence has shown that the acute effects of moderate-intensity aerobic exercise positively affect cognitive function [5,18].

### 1.1. High-Intensity Exercise

High-intensity exercise has been demonstrated to generate significant alterations in brain metabolism [13,19], resulting in higher levels of neurochemicals [16,19,20] that are hypothesized to negatively affect cognitive functioning. A decline in performance following high-intensity exercise has also been observed in other studies [14,21]. For instance, Mekari et al. (2015) [13] used a modified Stroop test to investigate information processing and executive function under low- (40 percent of peak power), moderate- (60 percent) and high-intensity conditions (85 percent). When comparing the high-intensity exercise group to the low-intensity group, they discovered a significant increase in reaction time (i.e., a slower response) and a significant decrease in accuracy. In contrast to cycling at 30 percent HRR, 50 percent HRR and at rest, Wang et al. (2013) [21] found substantial declines in executive function measures, as measured by the Wisconsin Card Sorting Test, during a cycling exercise at 80 percent HRR. In addition, Smith et al. (2016) [14] found that jogging on a treadmill at high effort (80% HRR) resulted in slower reaction times, mistake rates, omission rates and choice errors compared to running at moderate intensity (70%) and during resting settings. The lack of the high-intensity exercise characteristics looked at and the need for additional context to fully understand the consequences of the research on such exercise, however, make these studies stand out [5,22,23].

### 1.2. High-Intensity Exercise with Overloads

Extensive research has also been conducted on overload exercise. The study by Anders et al. (2021) [24] found that high-intensity overload exercise resulted in a very varied response pattern in cognitive processes, suggesting that there was a greater performance in cognitive domains related to fundamental computational skills than in cognitive domains related to memory and recall. This demonstrated response suppression suggests that high-intensity endurance exercise may induce distinctive differential responses in a number of cognitive domains. More and more data point to the possibility that exercise-induced alterations may be domain-specific [22,25,26,27].

For instance, Audiffren et al. (2008, 2009) [25,26] investigated the effects of cycloergometer exercise performed for 35 min at 90% of the ventilatory threshold and also showed significant improvements in reaction time throughout the exercise protocol [25]. Additionally, Chang et al. (2017) [13] used a Stroop test to measure executive function (response inhibition) and reported significant improvements in reaction time compared to the control group, but they did not observe any differences in reaction time for the incongruent task. In a fairly detailed meta-analysis on the short-term impact of exercise on cognitive performance, Wilke et al. [27] came to the conclusion that overload exercise appears to be a suitable strategy for the short-term improvement of cognitive function in healthy persons. The research shows that a single session of overload exercise produces moderate improvements in cognitive function compared to a control group that does not exercise, and the acute effects of resistance exercise are not superior to those following aerobic exercise.

### 1.3. Moderate/Low-Intensity Exercise

Acute aerobic exercise has been found to enhance blood flow in the brain at the level of moderate- or low-intensity exercise [28,29]. The neural demand, cardiac output and arterial carbon dioxide partial pressure are the primary regulators of cerebral perfusion following aerobic exercise, and it has been hypothesized that, during high-intensity overloaded exercise, variations in the blood flow occur through oscillations or spikes in arterial pressure [29]. Changes in the serum cortisol levels are a different potential contributor. After moderate-intensity overload exercise, higher levels of the stress hormone were detected in a study [16].

The subject of whether aerobic exercise might be slightly more effective than high-intensity training requires further investigation. Acute enhancements in cognitive performance may be beneficial in certain situations. Because athletes must integrate and process a wealth of sensory data at the supraspinal level in the majority of potentially traumatic scenarios, generating and revising motor plans under strict time restrictions, it may be crucial in the prevention of sports injuries. The key components of this paradigm are the cognitive domains examined—particularly, inhibitory control and cognitive flexibility [30]. A study by Wilkerson (2012) [31] discovered that neurocognitive reaction time might be utilized to predict lower extremity injuries, even though data from prospective trials are still lacking. For instance, just 2 of the 12 papers that made up Wilke et al. (2015)’s [27] meta-analysis included a further follow-up measurement. Pontifex et al. (2009) [32] showed no improvements in cognitive performance at 30 min after overload training in the first of these studies. The second experiment, conducted by Johnson et al. (2016) [33], discovered consistent but non-significant increases in various outcomes at 30 and 60 min. This was attributed to the high level of data variability. 

## 2. Materials and Methods

### 2.1. Study Design and Participants

The experiment involved two variables: type of exercise performed (HIFT or MICT) and cognitive performance. A randomized single-blind clinical trial was conducted with a control group (CTRL), an experimental group with high-intensity functional training (HIFT) and a group with moderate-intensity continuous training (MICT), in which a pre-intervention-posttest design will be used. The target population is young people between 18 and 25 years. The inclusion criteria for participation in the study were that the participants had an age within the target range and that the participants did not have any pathology that could influence the practice of physical exercise or physical activity. The recruitment of the sample was carried out by direct contact with university students in various higher education centers.

### 2.2. Sample Size Calculation

The sample size was determined with G*Power software (Version 3.1.9.7, Axel Buchner, Heinrich-Heine-Universität Düsseldorf, Düsseldorf, Germany). A priori analysis for a 3 [groups] × 2 [time] ANOVA model was performed for the main effect of time (i.e., pre, post) and intervention type (i.e., HIFT vs. MICT vs. CTRL), with an alpha level of 0.05 and a power of 0.80. Estimating an overall intervention effect of F *=* 0.35 and a correlation between repeated measures of 0.90, a total of 67 subjects in all (i.e., 22–23 subjects per group) would be necessary. Taking into account the above and considering a possible experimental mortality of 15%, at least 77 subjects (i.e., 25–26 per group) should be included in the study to find significant differences in the main variables of interest of this study. A total of 69 participants were recruited and divided between the different groups.

### 2.3. Allocation to Intervention

Three groups were defined in the study. A control group (CTRL) of 14 participants, an experimental group with high-intensity interval functional training (HIFT) of 27 participants and another experimental group with moderate-intensity continuous training (MICT) of 28 participants. The assignment to the groups was simple and involved concealed randomization. Those responsible for admitting the patients to the intervention phase did not know to which group they had been assigned. This assignment was made beforehand by a researcher who did not intervene in the subsequent phases of evaluation, intervention, data recording and database preparation. All the measurements described above were performed on the control group, the high-intensity group and the continuous training group just before the start of the intervention, and immediately afterwards, the results were recorded in a data log. The study was conducted in accordance with the Declaration of Helsinki and approved by the Ethics Committee of EADE University (005/PE/TS/2022) on 10 March 2022.

### 2.4. Procedure

The control group was not subjected to any training protocol but was evaluated in the pre- and post-study phase. After an initial evaluation, the HIFT group was subjected to a physical training session based on a circuit of 6 exercises (Squat, Push-Up, Lunge, Push-Press, Box Jump, Planks) for 30 min, performing 10 repetitions of each exercise continuously and resting 2 min each time they completed a round of the 6 exercises. Once the intervention was finished, they were again evaluated to see if there were differences with the results obtained in the initial evaluation. During training, the heart rate of all participants was monitored to maintain it above 85% of the maximum heart rate (HR). Finally, the moderate-intensity continuous training group (MICT), after an initial evaluation, was subjected to a physical training session on a cycloergometer for 30 min in which the heart rate was maintained between 70% and 80% of the heart rate maximum (HRM). Once the intervention was over, they were again subjected to a final evaluation (Figure 1).

### 2.5. Instruments

They were assessed with the Word Recall Test (verbal learning, delayed declarative memory), which consists of a series of memory trials with a list of 10 words. The individual is shown the words on the list and is asked to recall them after performing another unrelated task or after a delay. The score is calculated by recording the number of words recalled in each of the four trials [34]. Stroop test (executive function, selective attention): a list of words is presented with colors that matched the word (congruent, e.g., the word ‘red’ presented in red) or with colors that did not match the word (incongruent, e.g., the word ‘red’ presented in blue) [35,36]. N-back Test (working memory): a string of letters is presented one at a time on a screen, and participants must identify whether each letter presented is the same as or different from the previously presented letter [37].

### 2.6. Data Analysis

Once the data were collected, they were analyzed with SPSS v.25 software (SPSS Inc., Chicago, IL, USA). A descriptive study of the categorization variables was carried out (Table 1). Subsequently, a repeated means comparison was performed with ANOVA, where no significant differences were found between the groups in any of the variables, however, when comparing the groups with themselves between the pre-test and post-tests with the Student’s *t*-test for related samples. 

## 3. Results

Very significant changes (*p <* 0.01) were evidenced in the time of the fastest response (HIFT MD *=* −1.14, *p <* 0.01, d *=* 0.9; MICT MD *=* −1.79, *p <* 0.01, d *=* 0.9) and in the mean response time (HIFT MD *=* −2.16, *p <* 0.01, d *=* 0.66; MICT MD *=* −3.07, *p <* 0.01, d *=* 0. 9) in the Stroop Test (executive function and selective attention). Significant changes in the number of correct answers (*p <* 0.05) were evidenced for both the HIFT (MD *=* 1.08, *p <* 0.05, d *=* 0.5) and MICT (MD *=* 1.54, *p <* 0.05, d *=* 0.5) groups (Figure 2). In the control group, there were no such changes (Table 2, Table 3 and Table 4).

## 4. Discussion

A growing body of research indicates that the development of training-induced brain adaptations may aid in the prevention or postponement of cognitive decline and neurodegenerative diseases; chronic exercise encourages synaptic plasticity, angiogenesis and neurogenesis [37,38,39] based on data from animal experiments. Additionally, human investigations have shown that, following several weeks of training, neurotrophic factor (BDNF) expression and hippocampus brain volume increase [40,41]. Although the majority of this study concentrates on interventions with continuous exercise over time, these stated exercise-induced alterations in the brain appear to result in increased cognitive performance.

The results obtained are in agreement with the evidence found by other authors. For example, Kim et al., in 2015 [42], found a non-significant increase in neurotrophic factors (BDNF, NGF, IGF-1) in university students after Taekwondo training with respect to the control group. However, in the Stroop test results, they were significantly different (*p* < 0.05) in the exercise group with respect to the control. These findings suggest that Taekwondo exercise training can enhance cognitive functions—particularly, selective attention and executive function. Regarding working memory, Van den Berg et al. [43] assessed cognitive performance, measured before and immediately after exercise, by varying the exercise duration time. They performed an attention test and the N-Back task to measure selective attention and working memory, respectively. There were no significant effects of exercise on selective attention (i.e., alertness, orienting or executive control) or working memory performance immediately after the exercise sessions. In addition, there were no differential effects of exercise duration. In summary, acute exercise lasting 10, 20 or 30 min did not improve but also did not impair the cognitive performance of young adolescents compared to a sedentary control group. On declarative memory, Kathryn’s 2008 study [44] also agrees with our results. The researchers evaluated the effects of a brief session of moderate exercise on executive function, short-term memory and long-term memory tests. Eighteen young adults (mean age 22.2 years, sd ± 1.6) performed a game-changing test, a Brown–Peterson test and a Word Recall test before and after 40 min of moderate aerobic exercise on an ergometer bicycle, along with two control groups. After the exercise, they found no increases in game changing or declarative memory, suggesting that exercise-induced arousal does not influence the executive function processes involved in working memory. 

Chronic high-intensity training has been speculated to trigger adult neurogenesis, which is supported by recent data. For example, when Yarrow et al. (2010) [45] examined the association between overload training and BDNF expression, they detected elevated serum levels of the substance immediately after training. Furthermore, after 5 weeks of intervention, exercise-induced increases in BDNF were even more pronounced, suggesting the importance of this type of stimuli [46]. Higher serum levels of insulin-like growth factor-1 (IGF-1), which is related to neurogenesis and synaptogenesis [47], were measured after a 12-month intervention of overload training. Finally, Best et al. (2015) demonstrated that 52 weeks of overload training reduced age-related white matter atrophy in older women [48].

The strengths of the present study include the assessment of acute effects of HIFT and MICT training methods on cognitive functions. It is a new approach that differs substantially from those commonly employed in the literature in relation to exercise, which normally assess cognitive responses to prolonged training. Some limitations must be acknowledged concerning this study. The participants are all university students of physical education sciences, so their physiological and functional adaptations to exercise are above those the average university student; this could have reduced the acute effect of exercise on cognitive responses. Additionally, we must to point out the limitations of the losses in the control group. Some of the participants only did the pre-test and not the post-test; others did both but made mistakes in one or more of the tests.

## 5. Conclusions

Based on the data obtained, high-intensity and moderate-intensity exercise seem to have acute effects on executive function and selective attention but not on declarative or working memory.

There is no significant evidence that the acute effects of both types of exercise are superior to rest, although the improvement of the means in the HIFT and MICT groups is greater than that of the control group.

## Figures and Tables

**Figure 1 ijerph-19-10608-f001:**
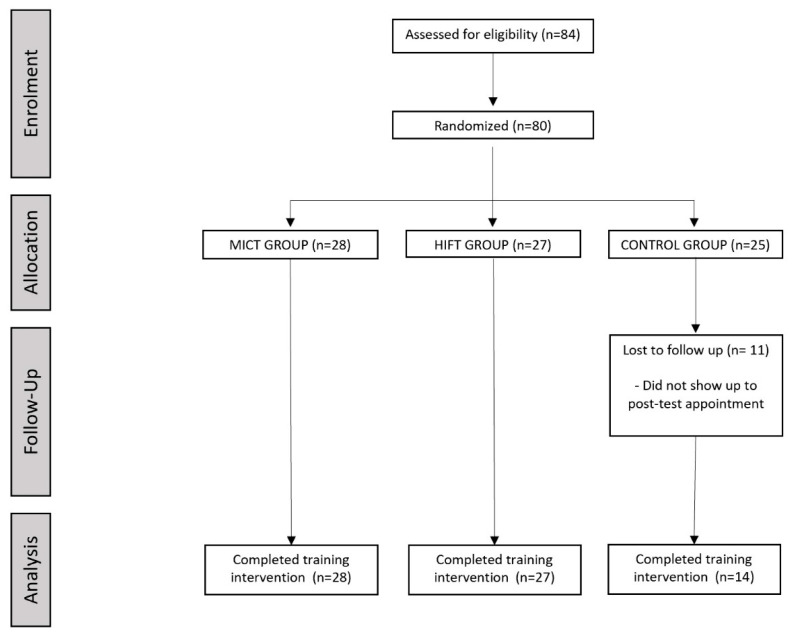
Flow Diagram of Study.

**Figure 2 ijerph-19-10608-f002:**
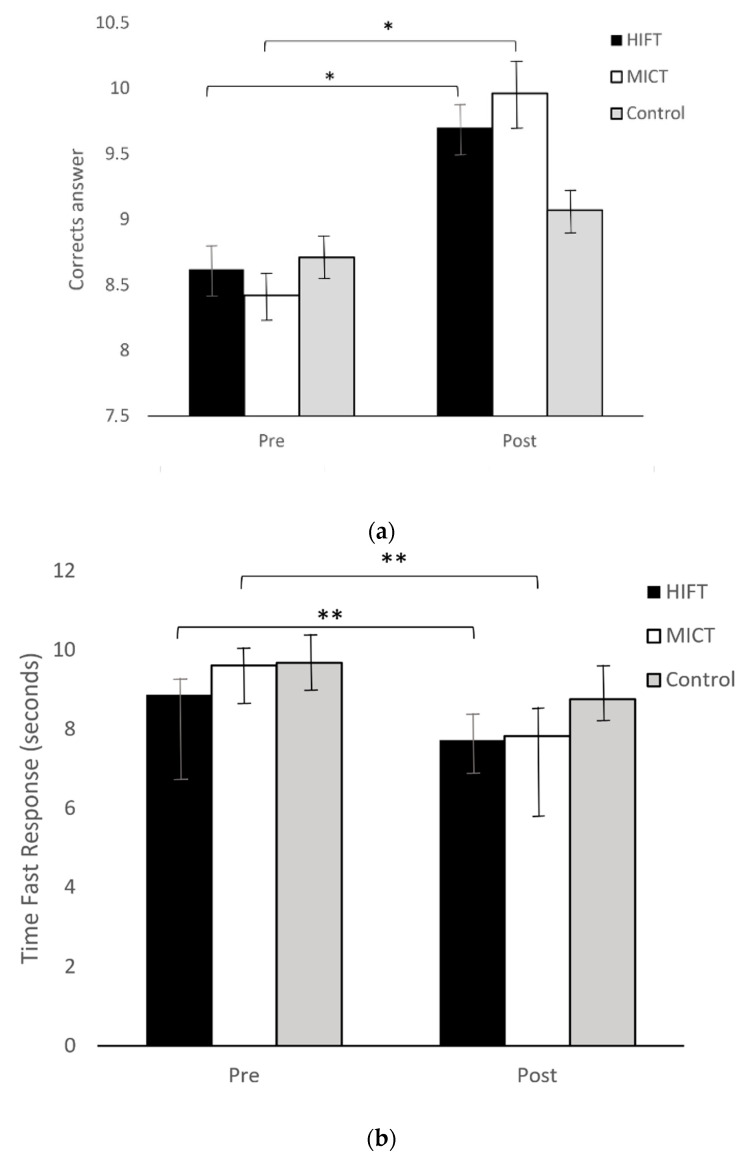
This figure shows the boxplot of the pre- and post-tests in the Stroop Test of each group in the three statistically significant variables. (**a**) Number of Correct Answers; (**b**) Fast Response Time; (**c**) Mean Response Time. * *p <* 0.05. ** *p <* 0.01.

**Table 1 ijerph-19-10608-t001:** Descriptives of age and IMC by group.

Group	N	Age (m ± sd)	IMC (m ± sd)
HIFT	27	21.62 ± 3.83	22.58 ± 1.97
MICT	28	20.25 ± 1.23	22.94 ± 0.80
CTRL	14	21.35 ± 2.43	22.93 ± 1.06

Data submitted as means ± standard deviation (m ± sd).

**Table 2 ijerph-19-10608-t002:** Control Group. Descriptives of pre-test, post-test and differences.

Variable		Pre-Test (m ± sd)	Post-Test (m ± sd)	Difference
Stroop	N° Corrects	8.71 ± 2.09	9.07 ± 3.42	0.36
	Fast Response (s)	9.68 ± 2.42	8.76 ± 2.32	−0.92
	Medium Response (s)	15.16 ± 3.77	13.47 ± 4.30	−1.69
Word Recall	% corrects	67.85 ± 12.51	67.14 ± 15.40	−0.71
N-Back	1-Back (% corrects)	91.9 ± 11.14	93.32 ± 8.67	1.42
	2-Back (% corrects)	80.47 ± 15.11	82.27 ± 16.40	1.8
	3-Back (% corrects)	76.17 ± 16.88	75.20 ± 15.58	−0.97

Data submitted as means ± standard deviation (m ± sd).

**Table 3 ijerph-19-10608-t003:** MICT Group. Descriptives of pre-test, post-test and differences.

Variable		Pre-Test (m ± sd)	Post-Test (m ± sd)	Difference
Stroop	N° Corrects	8.42 ± 2.58	9.96 ± 2.83	1.54 *
	Fast Response (s)	9.62 ± 2.17	7.83 ± 1.76	−1.79 **
	Medium Response (s)	14.20 ± 3.08	11.13 ± 2.03	−3.07 **
Word Recall	% corrects	67.14 ± 13.29	66.78 ± 13.34	−0.36
N-Back	1-Back (% corrects)	93.56 ± 10.65	93.08 ± 8.41	−0.48
	2-Back (% corrects)	82.64 ± 12.30	88.09 ± 9.47	5.45 *
	3-Back (% corrects)	72.17 ± 15.40	75.00 ± 14.47	2.83

Data submitted as means ± standard deviation (m ± sd). * *p <* 0.05. ** *p <* 0.01.

**Table 4 ijerph-19-10608-t004:** HIFT Group. Descriptives of pre-test, post-test and differences.

Variable		Pre-Test (m ± sd)	Post-Test (m ± sd)	Difference
Stroop	N° Corrects	8.62 ± 2.35	9.70 ± 2.38	1.08 *
	Fast Response (s)	8.87 ± 1.70	7.73 ± 1.57	−1.14 **
	Medium Response (s)	14.02 ± 3.48	11.86 ± 2.77	−2.16 **
Word Recall	% corrects	64.07 ± 13.08	68.51 ± 10.99	4.4
N-Back	1-Back (% corrects)	91.35 ± 11.37	90.87 ± 12.68	−0.48
	2-Back (% corrects)	77.80 ± 13.18	80.48 ± 14.89	2.68
	3-Back (% corrects)	72.10 ± 13.29	71.60 ± 13.78	−0.50

Data submitted as means ± standard deviation (m ± sd). * *p <* 0.05. ** *p <* 0.01.

## Data Availability

The data shown in this study are available upon request from the corresponding author. The data are not available to the public given the sensitive nature of the questions asked in this study and the necessary guarantees of privacy and confidentiality.

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
