# Peer review of "Acute Effects of High-Intensity Functional Training and Moderate-Intensity Continuous Training on Cognitive Functions in Young Adults"

_ijerph, 2022, doi:10.3390/ijerph191710608_

Round 1
Reviewer 1 Report
Authors Diego-Moreno et.al, have well written the manuscript entitled " Acute Effects of High Intensity Functional Training and Moderate Intensity Continuous Training on Cognitive Functions in Young Adults". Authors have performed the experiments in appropriate scientific methods and demonstrated the results well. However, reviewer has some minor concerns which needs to be addressed before publication.
1. The quality of graphs shown in Figure 2 are not up to the scientific publication level. Authors should improve the graphs. There is no Y axis title given. Please correct all graphs.
2. The introduction part of the manuscript is too lengthy and verbose. Please summerize the previous research in relation to the current research.
Reviewer 2 Report
First of all, I want to note that it has been a pleasure review your manuscript. I think this is an interesting topic.
This study assesses the acute effect of a HIFT and MICT training methods on the cognitive functions
In order to improve the quality of the manuscript. After reading in depth the manuscript, I would like to make some comments and ask the authors several questions about.
The introduction is too long and the discussion too short.
The introduction should be summarised by providing the most relevant data. Part of the introduction could be used for the discussion.
I would start the discussion with a sentence that serves as an introduction and not by talking directly about the results.
I would expand the section on limitations. For example, you could talk about the losses in the control group.
Round 2
Reviewer 1 Report
I am satisfied with with author's reply to comments. Authors have improved the graphs quality.